# Plcz1 Deficiency Decreased Fertility in Male Mice Which Is Associated with Sperm Quality Decline and Abnormal Cytoskeleton in Epididymis

**DOI:** 10.3390/ijms24010314

**Published:** 2022-12-24

**Authors:** Tao Wang, Binbin Cao, Yao Cai, Si Chen, Baozhu Wang, Yan Yuan, Quan Zhang

**Affiliations:** 1Institute of Comparative Medicine, College of Veterinary Medicine, Yangzhou University, Yangzhou 225012, China; 2Jiangsu Co-Innovation Center for Prevention and Control of Important Animal Infectious Diseases and Zoonoses, Yangzhou University, Yangzhou 225012, China

**Keywords:** Plcz1, CRISPR–Cas9, cytoskeleton damage, sperm quality, fertilisation, RNA sequencing

## Abstract

Phospholipase C zeta1 (Plcz1) was known to be a physiological factor in sperm that activates oocytes to complete meiosis by triggering Ca^2+^ oscillations after fertilisation. However, the role of male Plcz1 in spermatogenesis and early embryo development in progeny has been controversial. Plcz1 knockout (*Plcz1*^−/−^) mouse model (*Plcz1*^m3^ and *Plcz1*^m5^) was generated by using the CRISPR-Cas9 system. The fertility of *Plcz1*^−/−^ mice was evaluated by analysing the number of offsprings, sperm quality, pathological changes in the testis and epididymis. RNA-seq and RT-PCR were performed to screen differentially expressed genes and signalling pathways related to fertility in *Plcz1*^−/−^ mice. Further mechanism was explored by using *Plcz1*^−/−^ cells. *Plcz1* knockout led to hypofertility in male mice. In particular, a significant time delay in development and polyspermy was found in eggs fertilized by both *Plcz1*^m3^ and *Plcz1*^m5^ sperm. Interestingly, a decline in sperm quality combined with pathological changes in epididymis was found in *Plcz1*^m3^ mice but not in *Plcz1*^m5^ mice. Notably, abnormal cytoskeleton appears in epididymis of *Plcz1*^m3^ mice and *Plcz1*^−/−^ cells. Cytoskeleton damage of epididymis is involved in fertility decline of males upon Plcz1 deficiency in this model.

## 1. Background

The incidence of infertility is rising as the global population, which is now considered to affect 8–12% of couples worldwide [1,2]. To date, several sperm-borne oocyte activation factors (SOAFs) have been found to play vital roles in fertilization [3,4,5,6]. Sperm-specific phospholipase C zeta1 (*Plcz1*) was discovered and has been established as the predominant sperm oocyte-activating factor responsible for the characteristic free calcium (Ca^2+^) oscillations observed during mammalian fertilization [7,8,9]. The imperative role of Plcz1 in oocyte activation has been revealed by the mutations throughout the gene that have been identified and directly linked with certain forms of male infertility due to oocyte activation deficiency in human [10,11,12]. Clinical studies showed that six novel mutations and one reported mutation in *Plcz1* were identified in five of 14 independent families characterised by fertilisation failure or poor fertilisation, and these mutations may be responsible for fertilisation failure in men exhibiting primary infertility [10]. In terms of enzyme function, Plcz1 promotes the hydrolysis of phosphatidylinositol 4,5-bisphosphate (PIP_2_) to inositol triphosphate (IP_3_) and diacylglycerol (DAG), which promotes the release of Ca^2+^ from the endoplasmic reticulum and in turn induces intracellular Ca^2+^ oscillation [9,13]. In the past decade, male mice with sperm *Plcz1* defects were found to exhibit malformed sperm and infertility [14,15]. Recently, several studies have indicated that *Plcz1* knockout causes spermatogenesis failure [16,17,18]. However, there are studies claiming that partial or whole exon deletion of *Plcz1* using the CRISPR/Cas9 system does not affect spermatogenesis [19,20]. Although these experimental approaches indicated that *Plcz1* defects are associated with spermatogenesis failure, they do not supplant the use of a targeted gene deletion model. Therefore, genetically modified animals are currently necessary for investigating the molecular and cellular features of spermatozoa. Establishment of *Plcz1*-deficient animal model has been regarded as a key step toward answering this question.

Spermatogenesis is carried out in the convoluted seminiferous tubules of the testis, beginning with the proliferation of diploid spermatogonia and ending with the production of haploid spermatozoa [21,22]. The phenotype of sperm maturation caused by testis-specific gene mutation can lead to decreased fertility of males [23]. Notably, Plcz1 is an important catalysing enzyme in slender sperm cells during sperm maturation [24]. A variety of defects can be observed in some patients with round head spermatozoa, such as sperm with damaged cytoskeletons and sperm that are unable to activate oocytes [25]. In addition, Plcz1 is located in the acrosome and postacrosome regions, and its localisation changes dynamically during capacitation and the acrosome reaction [26], indicating that in addition to oocyte stimulation, sperm quality can also be affected by Plcz1 function. Despite the above findings, genetic evidence for the association between Plcz1 and fertilisation failure is still limited, and the genetic basis of fertilisation failure caused by Plcz1 mutations, especially in reproductive organs, requires further investigation.

In this study, *Plcz1* was knocked out in C57BL/6 mice by using the CRISPR–Cas9 system to explore the mechanism underlying the pathogenesis of reproductive disorders caused by *Plcz1* deletion and to provide new evidence for the occurrence of related diseases caused by *Plcz1* defects.

## 2. Results

### 2.1. Preparation of Plcz1 Knockout Mice

CRISPR-Cas9 gene editing that targeted two different conservative exons of *Plcz1* with independent single guide RNAs (sgRNAs) was employed to control for potential off-target mutagenesis (Figure 1A). Positive plasmids containing *Plcz1*-sgRNA6 and *Plcz1*-sgRNA7 were identified by DNA sequencing analysis. After that, 82 zygotes were obtained from donor mice, of which 54 live embryos were microinjected with the recombinant vector. Six offspring were obtained and numbered as 1–6. After amplification of the target fragment with PCR using primers for sgRNA6 and sgRNA7 detection, agarose gel electrophoresis was performed (Figure 1B). The results showed that there was an obvious band of 289 bp of mouse #3 (*Plcz1*^m3^), but no target band. However, a band of 268 bp was observed after amplified with sgRNA6 F and sgRNA7 R in *Plcz1*^m3^, which was shown to be 3346 bp in Wild Type (WT). Furthermore, the target sequences in the mice were analysed by DNA sequencing. As shown in Figure 1C, a 3078 bp deletion was found between exon 6 and exon 7 in *Plcz1*^m3^, while there was a 7 bp deletion in exon 7 of *Plcz1*^m4^. Regrettably, *Plcz1*^m4^ died during feeding. Furthermore, a 1 bp deletion was found in exon 6 of *Plcz1*^m5^. Further, the deletion of *Plcz1* was identified by western blot. As shown in Figure 1D, there was no obvious band of Plcz1 protein expression in *Plcz1*^m3^, but obvious bands can still be observed in *Plcz1*^m5^.

### 2.2. Decreased Fertility Was Caused by Plcz1 Knockout in Male Mice

To evaluate the fertility of *Plcz1*^−/−^ male mice, *Plcz1*^m3^ and *Plcz1*^m5^ were crossed with mice of different genotypes. Statistical analysis of the number of litters and litter size were performed after 13 weeks. As shown in Table 1 and Table 2, offsprings can be produced by *Plcz1*^−/−^ males, but at a reduced efficiency compared with *Plcz1*^+/+^ males. Specifically, the litter size produced by *Plcz1*^−/−^ males was significantly decreased compared with WT crosses (*Plcz1*^m3^: 2.5 ± 0.5; *Plcz1*^m5^: 2.3 ± 1.1; WT: 8.5 ± 1.4) with a decreased number of litters. These results indicate that the natural mating of both *Plcz1*^m3^ and *Plcz1*^m5^ male mice can produce offspring but with low fertility.

### 2.3. Abnormal Development of Eggs Was Aggravated by Plcz1-Null Sperm Fertilisation

To explore the effect of *Plcz1* deletion on early embryonic development in offspring, embryonic development in zygotes was observed under a microscope at different times after both in vivo and in vitro fertilisation. As shown in Figure 2A, the 2-cell embryo development rate of *Plcz1*^m3^ mice decreased significantly (*p* < 0.05) compared with WT after in vivo fertilization and in vitro culture (IVC); however, there was no significant difference in the 2-cell embryo development rate of *Plcz1*^m5^ mice at 48 h post-human chorionic gonadotropin (hCG) treatment. In addition, the morula development rate of *Plcz1*^m3^ and *Plcz1*^m5^ mice decreased significantly at 96 h post-hCG treatment (*p* < 0.01). Furthermore, the blastocyst development rates of *Plcz1*^m3^ and *Plcz1*^m5^ mice decreased significantly 120 h post-hCG treatment (*p* < 0.05). Compared with that of WT mice, the abnormal embryo rate of *Plcz1*^m3^ mice was significantly higher 48 h post-hCG treatment (*p* < 0.05), especially 120 h post-hCG treatment (*p* < 0.01). The abnormal embryo rate of *Plcz1*^m5^ mice was significantly higher at 96 h post-hCG treatment (*p* < 0.01). All of the details are shown in Appendix A.

Similarly, the result of in vitro fertilization (IVF) assay showed that both morula and blastocyst development rates of *Plcz1*^m3^ and *Plcz1*^m5^ mice decreased significantly (*p* < 0.05) at 96 h post-hCG treatment. Furthermore, the abnormal embryo rate of *Plcz1*^m3^ and *Plcz1*^m5^ mice was significantly higher than WT group by 120 h post-hCG treatment (*p* < 0.05) (Figure 2A and Appendix A). All these results indicate that the development of embryos can be delayed in eggs after fertilized with sperm from both *Plcz1*^m3^ and *Plcz1*^m5^ mice.

### 2.4. Polyspermy Was Increased after Fertilisation by Sperm from Plcz1^−/−^ Mice

It has been reported that polyspermy is one of the typical characteristics in eggs fertilised by Plcz1-null sperm. To identify whether polyspermy occurs, in this study, the number of PN in eggs fertilised by sperm from *Plcz1*^−/−^ mice was observed under a microscope. As shown in Figure 2B, most of the eggs fertilised by sperm from WT mice showed monospermy with an obvious 2PN stage (70/83). However, only 15/127 and 23/88 oocytes formed a 2PN after fertilisation with sperm from *Plcz1*^m3^ and *Plcz1*^m5^ mice. Most of the eggs showed failure of activation (unfertilised oocytes, *Plcz1*^m3^: 78/127; *Plcz1*^m5^: 37/88), abnormal activation (1PN, *Plcz1*^m3^: 19/127; *Plcz1*^m5^: 15/88) or polyspermic fertilisation (≥2PN, *Plcz1*^m3^: 15/127; *Plcz1*^m5^: 13/88). Additionally, the number of zygote pronuclei in *Plcz1*^−/−^ male mice was analysed under a laser confocal microscope after labelling with DAPI. As shown in Figure 2C, eggs fertilised by sperm from *Plcz1*^m3^ and *Plcz1*^m5^ mice displayed more than 2 pronuclei. All of these results suggest that deletion of the *Plcz1* leads to polyspermy and abnormal activation of oocytes.

### 2.5. Decline of Sperm Quality in Plcz1^m3^ Mice

To explore the effect of Plcz1 dysfunction on sperm motility and kinematic parameters, semen smears from *Plcz1*^−/−^ mice were collected and observed under a phase contrast microscope. As shown in Figure 3A, the sperm of *Plcz1*^m3^ appeared to be scarce and inactive compared with WT sperm. However, there were no significant differences in these parameters between *Plcz1*^m5^ and WT. Moreover, sperm motility was evaluated by using typical methods. As shown in (Appendix A), the proportion of forward motile sperm in *Plcz1*^m3^ was significantly lower (*p* < 0.01), and the proportion of inactive sperm was significantly higher (*p* < 0.01) than that in WT. In contrast, there was no significant change in sperm motility parameters in *Plcz1*^m5^. Similarly, the sperm velocity parameters (VCL, VSL, VAP) and ALH of sperm spatial displacement parameters of *Plcz1*^m3^ were much decreased (*p* < 0.05) (Figure 3B). However, there are no obvious changes of lipids or glucose in the plasma of *Plcz1*^m3^ and *Plcz1*^m5^ (Figure 3C). All these results indicate that loss of Plcz1 leads to a decrease in sperm quality in *Plcz1*^m3^ but not in *Plcz1*^m5^.

Furthermore, histological analysis of the testis and epididymis was performed by tissue section staining with HE. As shown in Figure 3D, the number of spermatogonia in the testis and spermatogenic cells of *Plcz1*^m3^ was greatly decreased, and the spermatogenic cells were exfoliated. However, there were no obvious pathological changes in *Plcz1*^m5^. In addition, a large number of round cells appeared in the lumen of epididymis (Figure 3E). Furthermore, epithelial cells in the luminal of the cauda epididymis was found to be shedding (Appendix A). These results indicate that Plcz1 knockout affects sperm quality, which is associated with pathological changes of epididymis in *Plcz1*^m3^ but not *Plcz1*^m5^.

### 2.6. Differential Expressed Genes in Epididymis of Plcz1^m3^ Mice

Epididymis is a highly convoluted duct lined by a pseudostratified epithelium that creates a unique luminal environment for sperm maturation. To clarify the mechanism underlying sperm maturation disorder caused by *Plcz1* knockout, RNA-seqencing (RNA-seq) was performed to screen the differential genes related to sperm maturation in the cauda epididymis of *Plcz1*^m3^ mice, which was verified by real-time fluorescence quantitative PCR.

A total of 518 differentially expressed genes (DEGs, 268 upregulated and 250 downregulated) were detected after *Plcz1* knockout using *p* < 0.05 as the cut-off value (Figure 4A,B). Based on the gene ontology (GO) categories, the identified DEGs were categorised into three major functional groups: supramolecular fibre, extracellular region, and antioxidant activity (Figure 4C), which are all involved in sperm maturation. To further investigate the functions of the DEGs, all DEGs were mapped to the Kyoto Encyclopedia of Genes and Genomes (KEGG) database. The following four pathways were significantly enriched (corrected *p* < 0.05): regulation of actin cytoskeleton, tight junction pathway, calcium signalling pathway and MAPK signalling pathway (Figure 4D). To verify the expression difference in related genes screened by RNA-seq, qPCR was used to detect the mRNA expression of sperm-related genes in the epididymis and testis of *Plcz1*^−/−^ mice. As shown in Figure 4E, the expression levels of *Ccdc7a*, *Spata18*, *Txndc2* and *Dkkl1,* which are involved in spermatogenesis, were significantly decreased in *Plcz1*^m3^ mice (*p* < 0.05) compared with WT mice. However, there was no significant change in the expression of *Prnd* and *Tssk2* (Figure 4E). In addition, the mRNA level of *Tubα3a*, a cytoskeleton-related gene, in the epididymis decreased significantly in *Plcz1*^m3^ mice compared with WT mice (Figure 4F).

### 2.7. Cytoskeleton Function Was Affected by Plcz1 Deletion in Epididymis

In order to explore the effect on of *Plcz1* deletion on the cytoskeleton in epididymis, frozen sections of cauda epididymis were prepared to observe the structures. F-actin and α-tubulin were labelled with fluorescence. As shown in Figure 5A and Appendix A, abnormal distributions of both F-actin and α-tubulin were found in the cauda epididymis of *Plcz1*^m3^. To further explore the effect, *Plcz1*^−/−^ cell line was prepared by using the CRISPR-Cas9 system. After labelling with fluorescent dye, cell filaments and microtubules were observed under a fluorescence microscope. As shown in Figure 5B, F-actin and a-tubulin stained as green or red filaments was distributed over the whole cell. However, both of the proteins mainly distributed around the plasma membrane to form a strong red fluorescence ring in the cell edge in *Plcz1*^−/−^ cells, which indicated that the structures of both actin and α-tubulin were impaired in cells with *Plcz1* knockout. Furthermore, the expression of microtubule motor proteins was detected by using western blotting. As shown in Figure 5C, the expression of motor protein chronophilin, cofilin, rac1/2/3 and Rho-Related GTP-Binding Protein RhoC was decreased with no effect on phosphorylation of MAPK and ERK in *Plcz1*^−/−^ cells compared with normal cells. Moreover, the phosphorylation of Ezrin, VASP and the expression of TESK1, which are related to cell mobility, were also inhibited by *Plcz1* deletion. These results indicated that Plcz1 plays an important role in maintaining the function of cytoskeletal dynamics, which is independent of MAPK signal pathway.

## 3. Discussion

In this framework, we demonstrated that *Plcz1* plays a vital role in the maintenance of reproductive capacity, sperm quality, as indicated by the declines in fertility and functional parameters including epididymal sperm motility and morphology in *Plcz1*-null male mice. Furthermore, we found that disorder of the cytoskeleton contributes to injuries of the testis and epididymis in mice with *Plcz1* knockout. Thus, spermatogenic failure is probably one of the key events underlying the subfertility caused by *Plcz1* mutation in male mice. This would make it possible to further assess the physiological significance of Plcz1. In addition, the mice model generated in this study would provide a null background in which to test the effects of Plcz1 of sperm maturation besides inducing egg activation and embryo development.

In contrast, there are previous articles reported that Plcz1-null mice were presented to be sterile with failure in oocyte activation but not sperm maturation [19]. Consistently, other studies have reported that *Plcz1* mutation leads to male sterility with blockade of spermatogenesis in mice; however, there are limitations to the techniques used to obtain *Plcz1* knockout mice [18]. In this study, CRISPR-Cas9 technology was used to produce *Plcz1*-deficient model while two genotypes of *Plcz1* gene knockout mice with frameshifts were generated, *Plcz1*^m3^ and *Plcz1*^m5^. Interestingly, Plcz1 protein could not be detected in the testis of *Plcz1*^m3^ mice but still be found in *Plcz1*^m5^ mice. We speculated that antigenic epitopes of Plcz1 in *Plcz1*^m5^ could be recognised by the commercial polyclonal antibody but missing in *Plcz1*^m3^. Recently, two different research groups have generated several *Plcz1* mutant mouse lines, which showed no problems in testis weight, spermatogenesis, or sperm swimming ability [1,2]. In this study, *Plcz1*^m3^ appeared to exhibit sperm maturation failure, with decreased testis weight and sperm quality, which appeared to be normal in *Plcz1*^m5^. We speculate that these discrepancies can be explained by the difference in the size of the missing fragment. Although the target site of sgRNA in the two mutants is identical, the mutations are completely different in this study. There is a deletion of 3078 bp in Plz1 gene of *Plcz1*^m3^ but only a single base deletion in *Plcz1*^m5^. As a member of PLC family, Plcz1 can hydrolyze PIP2 to produce not only IP3 but also diacyl glycerol (DAG) in cells. DAG is one of the best characterized products of PLC mediated reactions and that is known to lead to downstream activation of serine/thereonine protein kinase C (PKC). Further, PKC signaling is also required for the protection against oxidant-induced cytoskeletal disruption [27]. In this study, the regulation of actin cytoskeleton is affected by Plcz1 deletion according to the KEGG results. Thus, we speculated that the deletion region of Plcz1 in *Plcz1*^m3^ can lead to the loss of its hydrolytic function and further caused the inhibition of PKC signaling, which further aggravated the the disruption of cytoskeleton. Therefore, the differences in phenotypes between *Plcz1*^m3^ and *Plcz1*^m5^ may be consequent from the degree of the mutation, which needs explored in our further studies.

Studies have demonstrated that males with Plcz1 protein deficiency develop reproductive disorders in the process of reproduction [10,28]. Here, we found that the blastocyst development rate of *Plcz1*^−/−^ mice were significantly decreased, while the proportion of abnormal fertilised eggs were significantly increased. The oocytes of *Plcz1*^−/−^ male mice showed failure of activation, abnormal activation and polyspermy after mating for 12 h, which was consistent with the results of Nozawa. Moreover, blocking the plasma membrane during polyspermy plays a more important role than blocking the zona pellucida in vivo. Therefore, the combined defects of activation failure and polyspermy may explain the reduction in fertility. Typically, sperm abnormalities are mainly related to alterations in the spermatogenic process, which is closely associated with the physiological state of the reproductive organs [29,30]. In this study, pathological changes were found in the testes and epididymis of *Plcz1*^m3^ male mice according to histological analysis combined with declined sperm quality, which indicated that Plcz1 can be involved in physiological homeostasis of these organs and spermatogenic process. Furthermore, other previous reports have already suggested the involvement of the Capza3 gene, adjacent to the Plcz1 locus in the mouse/human genome, whose alteration cause spermatogenesis failure by single genetic alteration/deletion, in the abnormal spermatogenesis of Plcz1 gene deficient patients [31]. However, the relationship between Plcz1 deficiency and Capza3 need to be further explored. Herein, the low fertility potential added to the high percentage of preimplantation loss indicated a possible failure of the fertilisation ability of epididymal spermatozoa from Plcz1-null mice. In addition, compared with *Plcz1*^m3^ mice, *Plcz1*^m5^ mice had a higher fertilisation rate, higher mortality rate and lower percentage of unfertilised oocytes, which indicated that the mechanisms of fertility reduction in *Plcz1*^m3^ and *Plcz1*^m5^ mice may be different. The fertilisation failure of *Plcz1*^m3^ was mainly attributed to sperm maturation arrest and a decrease in sperm quality, which caused subfertility. In contrast, the decline in fertility in *Plcz1*^m5^ mice may be caused by a failure of egg activation and polyspermy. However, there are still many questions about the exact mechanism of Plcz1 and its role in the process of oocyte activation, and these issues need to be further explored.

Since the epididymis is the organ responsible for sperm maturation [32,33], in this study, RNA-seq was used to explore the effect of *Plcz1* deletion on epididymis physiology and the mechanisms underlying this process. Regarding KEGG pathway analysis, terms related to the regulation of the actin cytoskeleton, tight junction pathway, and calcium signalling pathway were found to be significantly affected by *Plcz1* knockout in the epididymis of *Plcz1*^m3^ mice. Furthermore, motor protein expression of the cytoskeleton was inhibited in TM3 cells. Actually, it has been suggested that actin binding proteins, in particular motor proteins play vital roles in spermatogenesis [34,35,36,37]. Thus, we speculated that Plcz1 plays a vital role in the maintenance of cytoskeletal motility in the reproductive organs of male mice. However, as RNA-seq is based on gene expression, multiple proteomics techniques are required to explore the specific mechanism in the future.

## 4. Materials and Methods

### 4.1. Administration of Mice

C57BL/6 and ICR mice were purchased from Yangzhou University Comparative Medical Center. The experiment was conducted in the laboratory of the Veterinary Building of College of Veterinary Medicine of Yangzhou University. All the experimental mice were raised in the clean animal room of Yangzhou University Comparative Medical Center. The light cycle was 5:00 a.m.–7:00 p.m. and the ambient temperature was controlled at 25 °C. The mice drank and ate freely, and the bedding was changed once a week.

### 4.2. Cell Culture

Cell culture and reagents TM3 were obtained from the American Type Culture Collection and cultured in DMEM/F12 media (Gibco; Thermo Fisher Scientific, Inc.) supplemented with 10% fetal bovine serum (Gibco; Thermo Fisher Scientific, Inc.), 100 U/mL penicillin and 100 µg/mL streptomycin in a humidified incubator containing 5% CO_2_ at 37 °C.

### 4.3. sgRNA Design

The information *Plcz1* gene sequence (NC_000072.7) of C57BL/6 mice were obtained from NCBI. The sequence of sgRNA targeting exons 6 and exons 7 of *Plcz1* were obtained from the website (http://crispr.mit.edu/, accessed on 10 September 2019) designed by Zhang Feng’ lab. Restriction site was added at both ends of the target sequences. The sequences of sgRNA were shown as following: *Plcz1*-sgRNA6 (F: CACCGAGATACACTACCG TCTCCAG, R: AAACCTGGAGACGGTAGTGTATCTC); *Plcz1*-sgRNA7 (F: CACC GCTTCCTATCACGGATCAAGG, R: AAACCCTTGATCCGTGATAGGAAGC). Then sgRNA was annealed to form double stranded.

### 4.4. Construction of Recombinant Vector

The annealed sgRNA double strands were inserted into pX330A vector in steps according to the in struction of Golden Gate method. The recombinant plasmid pX330A-*Plcz1* sgRNA6-*Plcz1* sgRNA7 was transformed into DH5α competent cells. After amplification, the specific plasmid was extracted by using Pure Plasmid Mini Kit.

### 4.5. Generation of Gene Knockout Mice

Male mice with vasectomy were prepared by removing the middle vas deferens and mated with the ICR female mice which had not been injected with hormone. After identification, the pseudopregnant female mice were obtained as receptors. Four-week-old C57BL/6 female mice were injected intraperitoneally with 8 IU of PMSG and 48 h later with 8 IU of hCG and were paired with C57BL/6 male mice. Zygotes were retrieved from oviductal ampullae at 22 h post-hCG. Zygotes surrounded by cumulus cells were digested with 2 mg/mL hyaluronidase in preheated H-CZB for 3 min, washed with fresh H-CZB and cultured in fresh balanced CZB. The plasmid (3 ng/μL) was injected into the pronucleus of the zygotes by microinjection. Before transplantation, the zygotes were transferred to the H-CZB culture medium and placed on the thermostat. The opening of the transplant tube containing zygotes was inserted into the oviduct infundibulum of recipient female mice for zygotes transplantation. During the transplantation, there were 15–20 embryos on each side of the fallopian tube. Under normal circumstances, the recipient female mice with successful transplantation can give birth to F0 generation mice after 3 weeks, and the delivery date and birth status are recorded. After sexual maturity, F0 generation mice were mated with wild-type C57BL/6 mice to obtain heterozygous genotypes.

### 4.6. Identification of Plcz1^−/−^ Mice by Using PCR

Genomic DNA was extracted from tail of the mice by digestion with proteinase K overnight at 55 °C with agitation in 0.5 mL lysis buffer [20 mM Tris-HCl (pH 8.0), 5 mM EDTA, 40 mM NaCl, 1% SDS, 0.4 mg/mL proteinase K]. The lysate was centrifuged at the maximum speed for 5 min at room temperature to obtain the upper aqueous phase containing DNA. Using the extracted mouse tail genomic DNA as a template, the target fragment containing the target sequence was amplified by PCR to verify the size of the target fragment. The detection primers for the corresponding fragments were shown as following: *Plcz1*^m3^ (F, TTAGAAAATCACTGCTCCCCTG; R, GAAGAGGAAGCTGACCCCTTAT); *Plcz1*^m4^(F, TATCTGAAACCCACGA GAGGAT; R, GAAGAGGAAGCTGACCCCTTAT); *Plcz1*^m5^ (F, TTAGAAAATCACTGCTCCCCTG; R, GCGTAGCAAA ACCATCTTCTCT). The obtained PCR amplification products were migrated in 2% agarose gel and analysed by DNA sequencing.

### 4.7. Western Blot

Protein samples were extracted from the testis tissue of mice and were lysed in RIPA buffer containing protease inhibitor PMSF. The supernatant was collected after 30 min on ice and centrifugation 15 min at 4 °C 12,000× *g*. The protein samples were mixed with 5 × SDS loading buffer and boiled for 10 min at 99 °C. After SDS-polyacrylamide gel electrophoresis, the protein was transferred onto PVDF membranes. The membranes were blocked with 5% skim milk in TBST for 2 h and thereafter incubated with primary antibody (anti-Plcz1; 1:1000, Thermo Fisher) at 4 °C overnight. After washing, the membranes were probed with secondary antibody conjugated to HRP (goat-rabbit IgG; CST) for 2 h.

### 4.8. Histology

The testis and epididymis were fixed in 4% paraformaldehyde, dehydrated with ethanol gradient and transparent with n-butanol and embedded in paraffin. For histopathological analysis, tissue samples were cut into 5 μm, and stained with hematoxylin and eosin for 8 min.

### 4.9. Sperm Morphology Analysis

Epididymis of Plcz1^−/−^ mice were placed in sperm capacitation solution incubated at 37 °C. The epididymal tails were isolated and incubated for 30 min to release the semen. The sperm were incubated with 5 μL suspension in the counting cell of Makler sperm counting plate and then observed with microscope of CASA automatic detection system. Each sample was randomly scanned for 4 to 5 visual fields. CASA automatically counted the sperm motility and kinematics parameters, including curvilinear velocity, straight-line velocity, average path velocity, straightness, linearity, wobble, amplitude of lateral head displacement, beat-cross frequency.

### 4.10. Fertility Assay

4-week-old C57BL/6 or F1 generation (B6×DBA) female mice were injected intraperitoneally with 8 IU of PMSG 48 h and later with 8 IU of hCG and were paired with *Plcz1*^−/−^ male mice. Zygotes were retrieved from oviductal ampullae at 22 h post-hCG, digested with 2 mg/mL hyaluronidase in preheated H-CZB for 3 min and washed with fresh H-CZB. According to the standard of storing 10 embryos in 20 μL CZB, all embryos were transferred to fresh balanced CZB and cultured in vitro in the presence of 5% CO_2_ at 37 °C. In addition, the male mice were killed to obtain the sperm of the male mice at 12–13 h post-hCG. Then the sperm was placed in the sperm capacitation fluid to capacitate for 1 h. Sperm capacitated for 40 min, cumulus oocytes were retrieved from oviductal ampullae, cultured in HTF. After 1 h of sperm capacification, 10 μL sperm suspensions were absorbed and added into HTF containing cumulus oocytes to fertilization for 5 h. Then all embryos were transferred to fresh balanced CZB and cultured in vitro in the presence of 5% CO_2_ at 37 °C. Development of embryos were observed and analysed every 24 h.

For litter assay, crosses were set up between males and females with genotypes described as WT, Het (Heterozygote), Homo (Homozygote) in *Plcz1*^m3^ and *Plcz1*^m5^ respectively. Pups born during the initial 13 weeks of mating from each pairs were counted respectively.

### 4.11. Fluorescent Staining

For tissue assay, samples were embedded in optimal cutting temperature compound (OCT) and rapidly frozen. Sections of 9-µm thickness were placed on poly-l-lysine-coated slides and fixed in cold acetone. Samples of tissue sections or cells were fixed in 4% paraformaldehyde solution for 30 min, 0.5% TritonX-100 was used to penetrate the membrane for 25 min at 25 °C. Then the samples were stained with actin-Tracker Green-488 (1:200, diluted by PBS) for 1 h. After that, samples were blocked with 5% BSA in PBS for 1 h and thereafter incubated with primary antibody (anti-alpha Tubulin, 1:1000, Abcam, Shanghai, China) at 4 °C overnight. After washed with PBS for 3 times, the samples were incubated with secondary antibody conjugated to rhodamine (1:200, Huabio, Hangzhou, China) for 1 h at room temperature. Finally, samples were stained with DAPI (1:1000, diluted by PBS) for 10 min at 25 °C and then were observed under laser confocal microscope for analysis.

### 4.12. RNA Preparation, Clustering, and Sequencing

Total RNA was extracted from the cauda epididymis of mice by using Trizol method. 3 μg of RNA per sample was used as input material for RNA sample preparations. Transcriptome libraries were generated using a NEBNext^®^ Ultra™ RNA Library Prep Kit for Illumina^®^ (NEB, Ipswich, MA, USA) following the manufacturer’s recommendations. Briefly, mRNAs were initially enriched with Oligod(T) beads. Enriched mRNAs were fragmented for 15 min at 94 °C. First strand and second strand cDNA were subsequently synthesized. cDNA fragments were end repaired and adenylated at 3′ends, and universal adapters were ligated to cDNA fragments, followed by index addition and library enrichment by PCR with limited cycles. The sequencing libraries were validated on the Agilent TapeStation (Agilent Technologies, Santa Clara, CA, USA), and quantified by using Qubit 2.0 Fluorometer (ThermoFisher Scientific) as well as by quantitative PCR. Final libraries were sequenced on a NovaSeq 6000 with Appendix A flow cell with 2 × 150 bp paired-end sequencing.

### 4.13. Determination of Blood Glucose and Blood Lipids

*Plcz1*^−/−^ mice were deprived of food one day before blood glucose test. Blood glucose was measured with the blood of *Plcz1*^−/−^ mice tail vein. The blood of the retroorbital venous plexus was used to detect blood lipids. After blood collection, disinfection and hemostasis were performed.

### 4.14. RNA Preparation and qRT-PCR

Total RNA was extracted from *Plcz1*^−/−^ mice testis and cauda epididymis by Trizol method. Then the total RNA was reverse transcribed by using HiScriptⅢ RT SuperMix for qPCR Kit (Takara, Dalian, China). Quantitative real-time PCR (qRT-PCR) was performed using ChamQ Universal SYBR qPCR Master Mix (Vazyme, Nanjing, China) on a StepOne instrument. Three parallel qRT-PCR responses are set to take their averages. The transcriptional level of β-actin was taken as the internal reference. All samples were run in the Applied Biosystems StepOne Real-Time PCR System (ABI, USA) and the 2^−ΔΔCt^ method was used to calculate the gene expression in different groups.

### 4.15. Statistical Analyses

IBM SPSS Statistics 22 or GraphPad prism 8 statistical software was used for statistical analysis. Student’s *t*-test was used for pairwise comparison between groups. Tukey-Kramer test for statistical multiplicity was also used in this study. The data were presented as average ± s.e.m. and the statistical charts were made by GraphPad Prism 7. Graphs are annotated with the following conventions: * *p* < 0.05, ** *p* < 0.01, *** *p* < 0.001.

## 5. Conclusions

Altogether, our study demonstrated that *Plcz1^–/–^* male mice can still conceive offspring in vivo but at a reduced efficiency, which is possibly associated with a decline in sperm quality and abnormal cytoskeleton in epididymis. Importantly, this study suggests a strategy to test the efficacy and safety of recombinant Plcz1 protein as an alternative therapeutic agent to treat infertility caused by *Plcz1* deficiency.

## Figures and Tables

**Figure 1 ijms-24-00314-f001:**
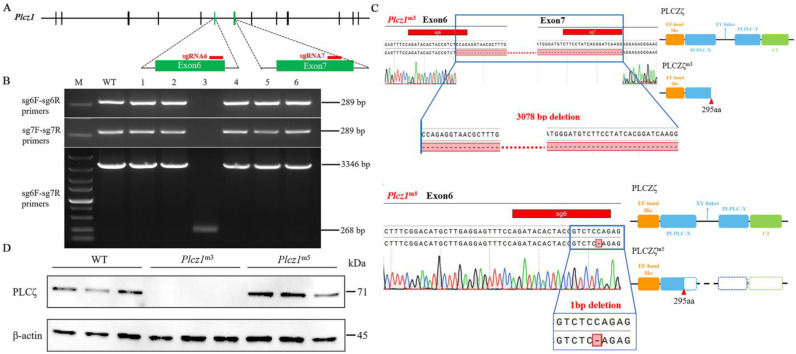
Recombinant vector sequencing and identification of *Plcz1*^−/−^mice. (**A**) Gene structure of the mouse *Plcz1* gene and target sequences (red lines) for CRISPR-Cas9. Exons are represented by vertical bars. (**B**) Identification of *Plcz1* gene mutation by using PCR: M, DL5000 marker; WT, wild type. (**C**) Comparison of genomic sequences from the WT allele and three mutant *Plcz1* alleles harbouring nucleotide deletions. The missing sequence is shown in red and marked with a dotted line. Predicted protein-domain structures for WT and truncated proteins resulting from the mutant *Plcz1*^m3^ and *Plcz1*^m5^. (**D**) Detection of PLCζ protein expression in the testes of *Plcz1*^m3^ and *Plcz1*^m5^.

**Figure 2 ijms-24-00314-f002:**
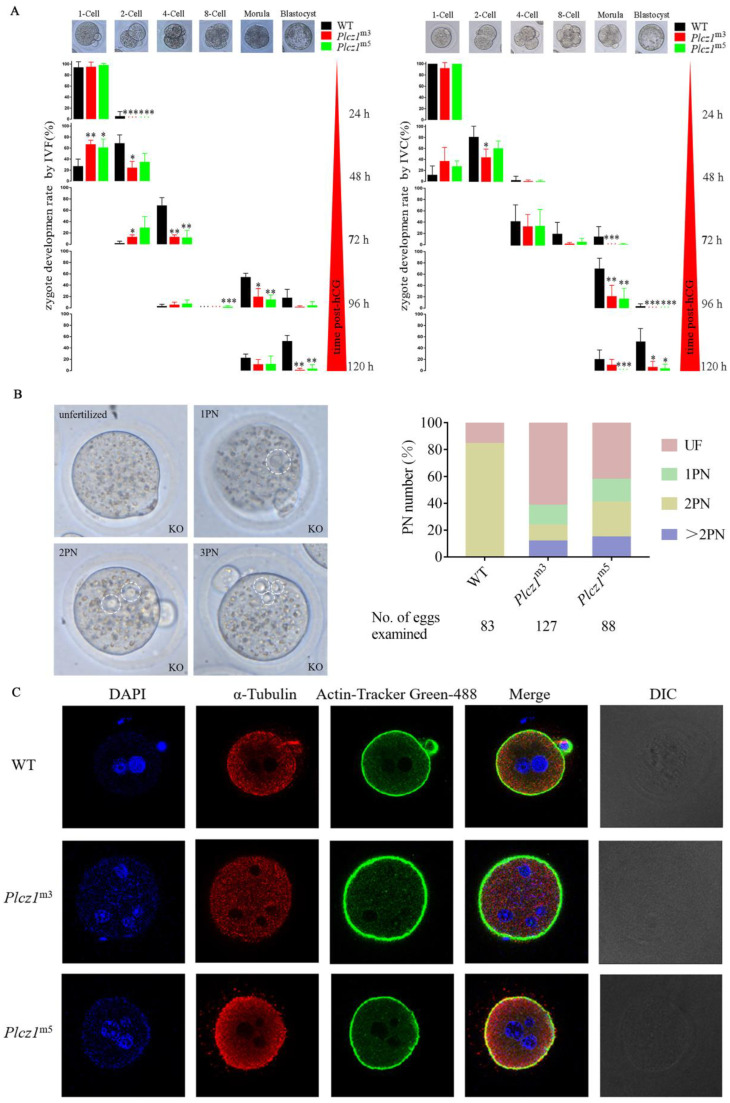
Delayed early embryonic development caused by sperm with *Plcz1* deletion is associated with polyspermy. (**A**) Early embryonic development of *Plcz1*^m5^ and *Plcz1*^m5^ offsprings after IVC or IVF in *Plcz1*^−/−^ mice compared with WT, * *p* < 0.05, ** *p* < 0.01, *** *p* < 0.001. (**B**) Oocytes with different numbers of pronuclei were analysed after observation under a microscope at 12 h post-coitus. Dotted circles indicate pronuclei. (**C**) After staining with DAPI, the number of pronuclei (PN) were observed in eggs fertilised with sperm of *Plcz1*^−/−^ mice under fluorescence microscopy (Scale bar = 20 μm).

**Figure 3 ijms-24-00314-f003:**
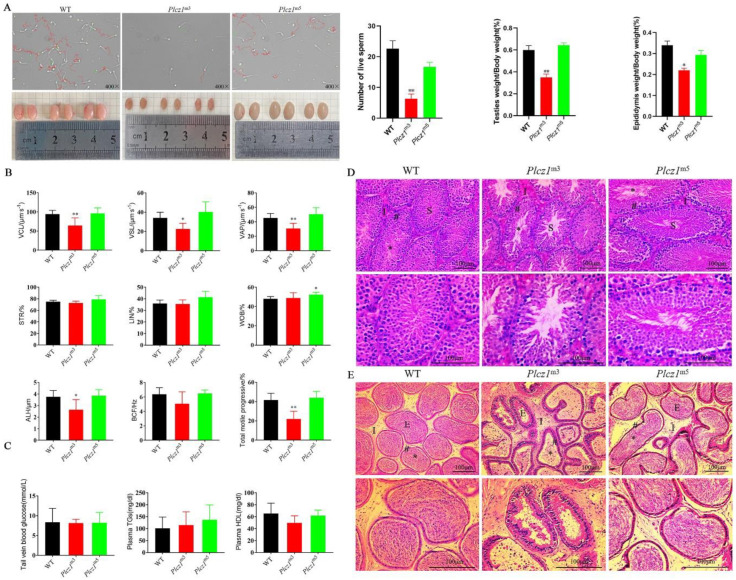
Characterisation of sperm and histopathological changes of testis and epididymis. (**A**) Morphological changes of sperm from *Plcz1*^m3^ and *Plcz1*^m5^ observed under a microscope: red indicates active sperm, green indicates inactive sperm, and yellow indicates impurities. The number of live sperms was statistically analysed by using GraphPad prism 8. In addition, the relative weight of testes and epididymis from *Plcz1*^m3^ and *Plcz1*^m5^ was measured (* *p* < 0.05, ** *p* < 0.01). (**B**) Sperm motility parameters were analysed with computer-assisted sperm analysis (CASA). VCL, curvilinear velocity; VSL, straight-line velocity; VAP, average path velocity; STR, straightness; LIN, linearity; WOB, wobble; ALH, amplitude of lateral head displacement; BCF, beat-cross frequency (* *p* < 0.05, ** *p* < 0.01). (**C**) Lipids and glucose assay were performed by using the kits. Histological analysis of (**D**) testes and (**E**) epididymis sections after stained with haematoxylin/eosin (HE). (For testes: S, seminiferous tubules; I, testicular stroma; *, seminiferous lumen; #, seminiferous tubule wall. For epididymis: E, epididymal canal; I, epididymal stroma; *, sperm mass; #, epididymal canal wall.

**Figure 4 ijms-24-00314-f004:**
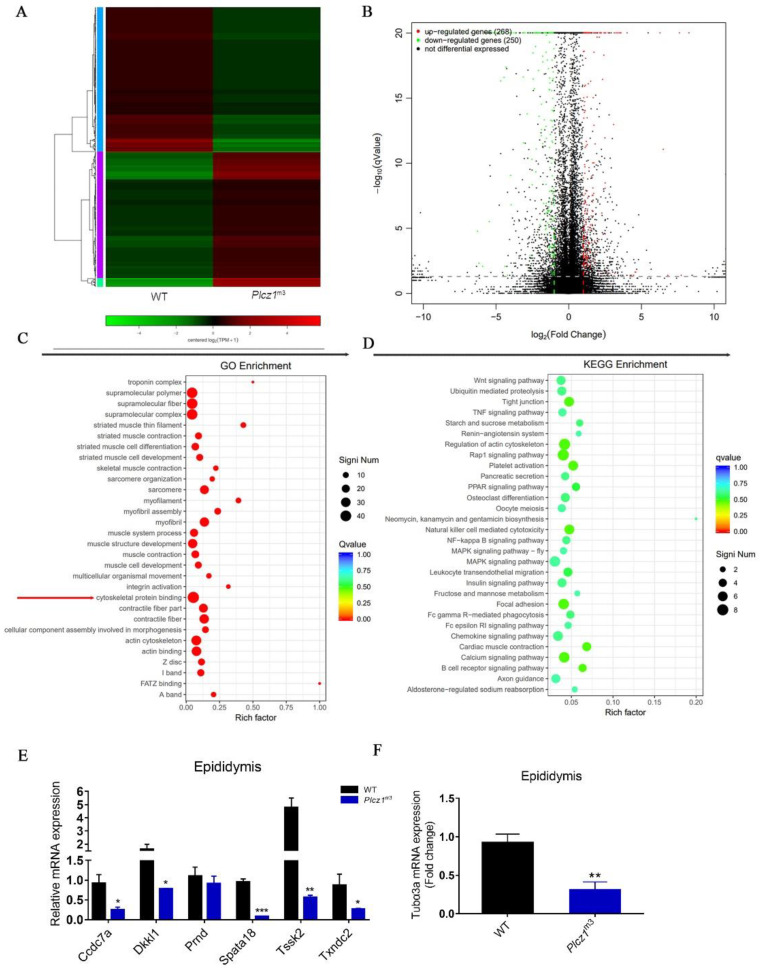
RNA sequencing analysis of cauda epididymis *Plcz1*^m3^ mice. (**A**) Hierarchical clustering analysis of z-scored expression profiles of transcripts that were up-regulated or down-regulated between *Plcz1*^m3^ and WT. (**B**) Volcano plot of unigenes in *Plcz1*^m3^ compared with WT. (**C**) Functional gene ontology categories of differentially expressed genes between *Plcz1*^m3^ and WT. (**D**) Scatter plot of differentially expressed genes enriched in KEGG pathways. The enrichment factor represents the ratio of the number of DEGs to the number of all the unigenes in the pathway; the *q* value represents the corrected *p* value. (**E**,**F**) Relative expression level of differentially expressed genes (*Ccdc7a*, *Dkkl1*, *Pmd*, *Spata18*, *Tssk2*, *Txndc2* and *Tuba3a*) in epididymis of *Plcz1*^m3^ were detected by using RT-PCR (* *p* < 0.05, ** *p* < 0.01, *** *p* < 0.001).

**Figure 5 ijms-24-00314-f005:**
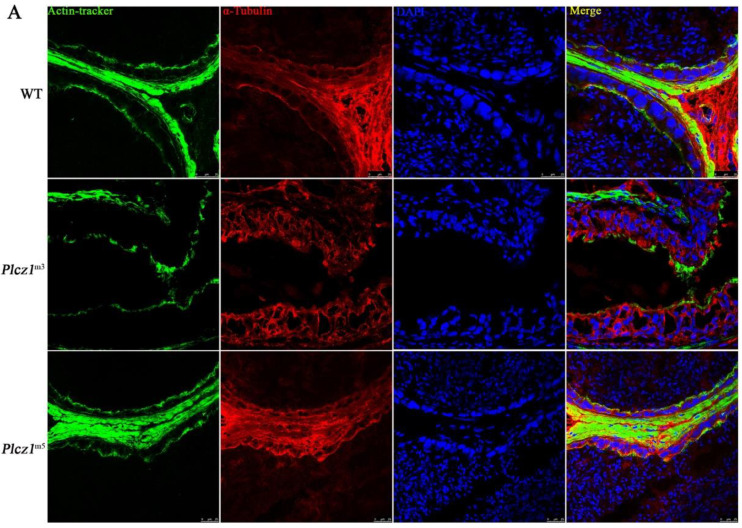
Effects of *Plcz1* deletion on cytoskeleton in cauda epididymis and TM3 cells. (**A**,**B**) DAPI, α-microtubule antibodies, Rodin Ming-Phalloidin were used to stain cell nuclei, microtubules, and microfilaments, respectively. The cytoskeleton of TM3 cells was observed under the confocal laser microscope (Scale bar = 10 μm). (**C**) The effect of *Plcz1* knockdown on the expression of cytoskeletal motor proteins (chronophilin, cofilin, rac1/2/3 and RhoC) in TM3 cells (* *p* < 0.05, ** *p* < 0.01).

**Table 1 ijms-24-00314-t001:** Fertility parameters of *Plcz1*^m3^ ($\bar{x\;}$ ± *s*, * *p* < 0.05, ** *p* < 0.01).

Crosses Male × Female	Number of Litters	$\mathrm{Litter}\;\mathrm{Size}\;(\bar{x}\;\pm \;s)$
WT(*n* = 6) × WT(*n* = 7)	10	8.5 ± 1.4
Het^(m3)^ (*n* = 16) × Het^(m3)^ (*n* = 15)	25	6.3 ± 1.5 *
Het^(m3)^ (*n* = 6) × Homo^(m3)^ (*n* = 7)	9	6.9 ± 1.5 *
Homo^(m3)^ (*n* = 15) × WT(*n* = 15)	2 **	2.5 ± 0.5 **
Homo^(m3)^ (*n* = 17) × Het^(m3)^ (*n* = 17)	3 **	3.0 ± 0.8 **
Homo^(m3)^ (*n* = 10) × Homo^(m3)^ (*n* = 6)	3 **	3.0 ± 2.2 **

**Table 2 ijms-24-00314-t002:** Fertility parameters of *Plcz1*^m5^ ($\bar{x\;}$ ± *s*, ** *p* < 0.01).

Crosses Male × Female	Number of Litters	$\mathrm{Litter}\;\mathrm{Size}\;(\bar{x\;}\;\pm \;s)$
WT(*n* = 6) × WT(*n* = 7)	10	8.5 ± 1.4
Het^(m5)^ (*n* = 10) × Het^(m5)^ (*n* = 11)	13	8.0 ± 2.3
Het^(m5)^ (*n* = 6) × Homo^(m5)^ (*n* = 4)	3	6.7 ± 1.3
Homo^(m5)^ (*n* = 13) × WT(*n* = 22)	11	2.3 ± 1.1 **
Homo^(m5)^ (*n* = 9) × Het^(m5)^ (*n* = 10)	2 **	2.5 ± 0.5 **
Homo^(m5)^ (*n* = 5) × Homo^(m5)^ (*n* = 5)	0 **	0 **

## Data Availability

The data presented in this study are available on request from the corresponding author.

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
