# Peer review of "Plcz1 Deficiency Decreased Fertility in Male Mice Which Is Associated with Sperm Quality Decline and Abnormal Cytoskeleton in Epididymis"

_ijms, 2022, doi:10.3390/ijms24010314_

Round 1

Reviewer 1 Report

The article by Wang et al describes the KO of the Plcz1 gene. Two different strains were obtained and phenotyped.

It is necessary to note that some of these results were already published by other authors. This paper tries to focus on the spermatogenesis angle.

Some general suggestions:

Discuss the mouse background effect?

Study calcium waves induced by these KO sperm in comparison to previous models?

The work seems sound but some imprecisions in the manuscript make it hard to read.

In addition, figures are too dense therefore compacted and unreadable without zooming. They should be arranged differently.

L41 : IP3 should be inositol triphosphate and not the repeat of PIP2

L75 : donor mice “after pregnancy”???

Figure 1 :

Panel A not necessary

Panel C : there is still some Chinese writing.

The sequence is not always readable

It is very surprising that the m3 and m5 mutants generate an identical truncated protein as depicted here. This should be checked and discussed in the text. Specially as the related phenotypes are different.

Table 1 and 2 : “Homo” rather than “Mut” if compared to “Het”

Mortality but at which stage?

Figure 2, panel A : Use more contrasting colors

L170 : the decrease is not obvious from a small picture. A quantification would be appreciated

Figure 3 :

Use more contrasting colors for histograms again.

There is a discrepancy between the figure and the legend : there is no F panel and the rest has been mixed up.

L204-210 : The authors seem to have mixed this paragraph with another article. Results here seem to depict a rat model of arsenic exposure…

L280 : I don’t understand the story about Capza3…

Author Response

Dear Reviwer,

Thank you very much for your review on our manuscript "Plcz1 deficiency decreased fertility in male mice which is attributed to cytoskeleton damage of germ cells" (ijms-2039573). These comments for the revision are really valuable and helpful for revising and improving our paper, as well as the important guiding significance to our researches.

We have tried our best to revise the manuscript according to your comments and have made revision which marked in red in the paper. Attached is the point-by-point response to all of your comments and suggestions. We would like to express our great appreciation to you for comments on our paper.

The main corrections in the paper and the responds to the reviewer comments point to point are as following,

Specific response to reviewer #1:

Question: Discuss the mouse background effect?

Response: Thanks for your valuable suggestion. We discussed more details about the mouse background effect in the “Discussion” with red color.

Question: Study calcium waves induced by these KO sperm in comparison to previous models?

Response: Thanks for your valuable suggestion. It is well known Ca2+ oscillations in eggs can be induced by Plcz1 after fertilization. Actually, measurement of calcium waves in zygotes of Plcz1m3 and Plcz1m5 has been performed in preliminary experiment of this study. We found that there is an obvious defect of calcium waves in zygotes fertilized by sperm from both genotypes compared with wild type, which is consistant with previous study. Here, are focus on the possible effect of Plcz1 defect on cytoskeleton.

Question: The work seems sound but some imprecisions in the manuscript make it hard to read. In addition, figures are too dense therefore compacted and unreadable without zooming. They should be arranged differently.

Response: Thanks for your kind reminding, and we have improved the precisions throughout the manuscript and made rearrangement of figures properly in the revision.

Question: L41: IP3 should be inositol triphosphate and not the repeat of PIP2

L75: donor mice “after pregnancy”???

Response: Thanks for your kind reminding. We have revised clerical errors in the revision.

Question: Figure 1:

Panel A not necessary

Panel C: there is still some Chinese writing.

The sequence is not always readable.

Response: Thanks for your valuable suggestion. We have deleted Panel A and revised the writing in Panel C. In addition, we rearranged the figures to make it readable according to your requirement.

Question: It is very surprising that the m3 and m5 mutants generate an identical truncated protein as depicted here. This should be checked and discussed in the text. Specially as the related phenotypes are different.

Response: It is a very good question. Actually, we are also puzzled about this. It is absolutely right that we could not determine the truncated protein just according to DNA sequencing analysis. According to the result of WB, the truncated proteins of Plzc1 are different. Although the target site of sgRNA in m3 and m5 mutants is identical, the mutations are completely different in this study. There is a deletion of 3078 bp in Plz1 gene of m3 but only a single base deletion in m5. Therefore, we speculated that the differences in phenotypes between m3 and m5 may be consequent from the degree of the mutation, which needs explored in our further studies. As you suggested, we have discussed the question in the disscusion of the revision.

Question: Table 1 and 2: “Homo” rather than “Mut” if compared to “Het”

Mortality but at which stage?

Response: Thanks for your kind reminding. We have changed “Mut” to “Homo” in the tables according to your suggestion. Mortality in the tables is referred to the offspring died at about 1-2 days due to objective factors. As the “Mortality” is not stressed in this study, so is not shown in the revision.

Question: Figure 2, panel A: Use more contrasting colors.

Response: Thanks for your valuable suggestion, and we used more contrasting colors in Figure 2, panel A as you suggested in the revision.

Question: L170: the decrease is not obvious from a small picture. A quantification would be appreciated.

Response: Thanks for your valuable suggestion, and we provided the quantification of the picture in Figure 3A of the revision.

Question: Figure 3, Use more contrasting colors for histograms again.

Response: Thanks for your kind reminding, and we used more contrasting colors in the revision according to your suggestion.

Question: Figure 3, There is a discrepancy between the figure and the legend: there is no F panel and the rest has been mixed up.

Response: Yes, we are so sorry for the clerical error, and we organised the panels properly in the revision.

Question: L204-210: The authors seem to have mixed this paragraph with another article. Results here seem to depict a rat model of arsenic exposure…

Response: Thanks for your kind reminding. Yes, it is improperly for us to introduce the related genes in this way. We changed the words in another way in the revision.

Question: L280: I don’t understand the story about Capza3…

Response: Thanks for your question. There was a with our statement about Capza3, and we have changed the statement into “Furthermore, other previous reports have already suggested the involvement of the Capza3 gene, adjacent to the Plcz1 locus in the mouse/human genome, whose mutation cause spermatogenesis failure by single genetic alteration/deletion, in the abnormal spermatogenesis of Plcz1 gene deficient patients” in the revision. Actually, Capza3 is adjacent to the Plcz1 locus in the mouse/human genome. Further, Capza3 mutation was reported to be involved in spermatogenesis failure, which may be caused by Plcz1mutation. Thus, it needs to be further explored about relationship between Plcz1 deficiency and Capza3.

Reviewer 2 Report

In this study, “Plcz1 deficiency decreased fertility in male mice which is attributed to cytoskeleton damage of germ cells,” the authors investigated Plcz1 knockout mice related to male reproductive abilities. Unfortunately, the reviewer could not understand the logic of the manuscript and what is novel because of the insufficient Materials and methods and low-quality images. Please consider the following comments.

Major comments:

  1. In the title, the authors focus on the cytoskeleton of germ cells, but in the Abstract, the authors declare that “the spermatogenesis disorder in Plcz1m3 mice is closely related to functional defects in cytoskeletal proteins in the epididymis.” The results for the abnormality of the cytoskeleton in germ cells are not shown in the manuscript. Furthermore, the reviewer wonders if the cytoskeleton of the epididymal epithelium is really affected. 
  2. The method and results for fertility assay are insufficient. How natural mating was set is lacking in the Materials and methods. Fertility parameters are shown in Tables 1 and 2, but the reviewer could not understand what parameter was obtained by in vivo or vitro assay. The authors need to clarify the results to show whether it was obtained from natural mating or in vitro fertilization. Does the fertilization rate mean the pregnancy rate? What is the difference between the number of litters and litter size? To discuss the reason for lower fertility in Plcz1 knockout mice, both in vivo and in vitro fertility assays are necessary.
  3. The authors performed a student’s t-test to compare groups, but statistical multiplicity was not considered. Some methods, i.e., Bonferroni’s p-value adjustment or Tukey-Kramer test, are necessary.
  4. The ductule shown in the magnified view of Fig. 3E (Plcz1m3) appears to be vas deferens. As described, the authors should prepare a picture to focus on the luminal components of the cauda epididymis. Fig. 3F is lacking.
  5. The Materials and methods for the RNA sequencing analysis are lacking. Furthermore, the obtained data needs to be placed in a public database. The reviewer wonders why the authors focused on the epididymis. Please show the histology of the epididymis in more detail because the epithelial cells should show abnormal structures if the cytoskeleton is abnormal. Confirmation by qPCR is also necessary.
  6. For abnormality in germ cells, high-quality images of the seminiferous tubule are needed. Furthermore, the stage of spermatogenesis should be the same to compare wild and mutants.
  7. The reviewer wonders why the phenotype is different between m3 and m5 mutants. The authors should clearly show the differences between these mutations and discuss how the differences cause the phenotype observed.

Minor comments:

  1. Chinese characters are in Fig. 1. 
  2. The authors should add an embedding medium (paraffin?) to the 4.8. Histology section.
  3. The authors should specify the “4% histiocytic fixative” used to fix fertilized eggs.
  4. The authors should define all the abbreviations at the time of the first appearance.
  5. L279–282 and L286–288 require references.

Author Response

Dear Reviwer,

Thank you very much for your review on our manuscript "Plcz1 deficiency decreased fertility in male mice which is attributed to cytoskeleton damage of germ cells" (ijms-2039573). These comments for the revision are really valuable and helpful for revising and improving our paper, as well as the important guiding significance to our researches.

We have tried our best to revise the manuscript according to your comments and have made revision which marked in red in the paper. Attached is the point-by-point response to all of your comments and suggestions. We would like to express our great appreciation to you for comments on our paper.

The main corrections in the paper and the responds to the reviewer comments point to point are as following,

Specific response to reviewer #2:

Question 1: In this study, “Plcz1 deficiency decreased fertility in male mice which is attributed to cytoskeleton damage of germ cells,” the authors investigated Plcz1 knockout mice related to male reproductive abilities. Unfortunately, the reviewer could not understand the logic of the manuscript and what is novel because of the insufficient Materials and methods and low-quality images.

  1. In the title, the authors focus on the cytoskeleton of germ cells, but in the Abstract, the authors declare that “the spermatogenesis disorder in Plcz1m3 mice is closely related to functional defects in cytoskeletal proteins in the epididymis.” The results for the abnormality of the cytoskeleton in germ cells are not shown in the manuscript. Furthermore, the reviewer wonders if the cytoskeleton of the epididymal epithelium is really affected.

Response: We are appreciated to your question. You are absolutely right about the confusing question in the manuscript. In this study, we prepared Plcz1 knockout mice to investigate the role of Plcz1in male reproductive abilities. The result showed that reproductive abilities was decreased in Plcz1 knockout mice in both genotypes. Then we want to know the possible mechanisms, as it is controversial. We found the quantity sperm is decreased with abnormal physiological parameter and pathological changes in epididymitis of Plcz1m3, which was rarely reported. As the epididymis is the organ responsible for sperm maturation, RNA-seq was used to explored possible mechanism. We found that the expressions of several key cytoskeleton related genes were changed in different degree. It is inaccuracy for us to use the statement of “cytoskeleton of germ cells” in the manuscript, and we modified the statement in the revision. Further, we tried our best to do supplementary trial and supply the evidence of abnormal cytoskeleton in epididymal epithelium in Figure 5 of the revision.

Question 2: (1) The method and results for fertility assay are insufficient. How natural mating was set is lacking in the “Materials and methods”.

Response: Thanks for your valuable suggestion. As you suggested, we supplied the natural mating method in the Materials and methods L377-L380.

(2) Fertility parameters are shown in Tables 1 and 2, but the reviewer could not understand what parameter was obtained by in vivo or vitro assay. The authors need to clarify the results to show whether it was obtained from natural mating or in vitro fertilization. To discuss the reason for lower fertility in Plcz1 knockout mice, both in vivo and in vitro fertility assays are necessary.

Response: Thanks for your question. Actually, fertility parameters shown in Tables 1 and 2 were both obtained by in vivo assay. Your question is valuable. It is a pity for us to miss the in vitro assay of the litter fertility parameters, and we supplied the data of early embryonic development by fertilization in vitro and in vivo in Table S1 and Table S2 respectively, which can be also viewed as a kind of fertility parameters on some way.

(3) Does the fertilization rate mean the pregnancy rate? What is the difference between the number of litters and litter size?

Response: Thanks for your question. In this study, fertilization rate is different from pregnancy rate. The number of litters means number of the litters produced by the pairs, while litter size is referred to the number of the offsprings in one litter in this study.

Question 3: The authors performed a student’s t-test to compare groups, but statistical multiplicity was not considered. Some methods, i.e., Bonferroni’s p-value adjustment or Tukey-Kramer test, are necessary.

Response: We are appreciated to your suggestion. Actually, Tukey-Kramer test for statistical multiplicity was also used in this study. We are sorry for the clerical error to miss it and we modified it in the revision.

Question 4: The ductule shown in the magnified view of Fig. 3E (Plcz1m3) appears to be vas deferens. As described, the authors should prepare a picture to focus on the luminal components of the cauda epididymis. Fig. 3F is lacking.

Response: Thanks for your valuable question. As you suggested, we supplied the picture that focus on the luminal components of the cauda epididymis in the Fig. S1 of supplementary material. We sorry for the lack of Fig. 3F and we modified the panels of Fig. 3 properly in the revision.

Question 5: The Materials and methods for the RNA sequencing analysis are lacking. Furthermore, the obtained data needs to be placed in a public database. The reviewer wonders why the authors focused on the epididymis. Please show the histology of the epididymis in more detail because the epithelial cells should show abnormal structures if the cytoskeleton is abnormal. Confirmation by qPCR is also necessary.

Response: Thanks for your kind reminding. We supplied the methods for the RNA sequencing analysis according to your requirement and we tried to place the obtained data in the public database. Recently, more and more studies revealed that bioactive components in sperm, such as proteins, RNA and epididymosomes can be obtained from epididymis, which is closely associated with the sperm quality and fertility of males are closely associated with gene expression in epididymis (James ER etal. The Role of the Epididymis and the Contribution of Epididymosomes to Mammalian Reproduction, Int J Mol Sci. 2020. Yu Liu etal, Journal of Agricultural and Food Chemistry 2019). Here, we found that Plcz1 deletion lead to decline of sperm quality and pathological change in epididymis. Thus, we want to know the possible mechanism by using RNA-seq analysis. In addition, we supplied confirmation of gene expression by qPCR in Fig. 4E and 4F. In addition, the cytoskeleton of the epididymis in more detail in Fig. 5A as you suggested.

Question 6: For abnormality in germ cells, high-quality images of the seminiferous tubule are needed. Furthermore, the stage of spermatogenesis should be the same to compare wild and mutants.

Response: We are appreciated to your suggestion. According to your reminding, we realized that the conclusion about sperm quality affected by Plcz1 can be determined in epididymis but not in germ cells. We have modified the relative statements in the revison. Your suggestion is very valuable for our further studies, thanks again.

Question 7: The reviewer wonders why the phenotype is different between m3 and m5 mutants. The authors should clearly show the differences between these mutations and discuss how the differences cause the phenotype observed.

Response: Thanks for your valuable question. It is absolutely right that we could not determine the truncated protein just according to DNA sequencing analysis. According to the result of WB, the truncated proteins of Plzc1 are different. Although the target site of sgRNA in m3 and m5 mutants is identical, the mutations are completely different in this study. There is a deletion of 3078 bp in Plz1 gene of m3 but only a single base deletion in m5. Therefore, we speculated that the differences in phenotypes between m3 and m5 may be consequent from the degree of the mutation, which needs explored in our further studies. As you suggested, we have discussed the question in the disscusion of the revision.

Minor comments:

  1. Chinese characters are in Fig. 1.

Response: Thanks for your reminding and we have modified it in the revision.

  1. The authors should add an embedding medium (paraffin?) to the 4.8. Histology section.

Response: Thanks for your reminding and we added the embedding medium to the 4.8. Histology section in the revision.

  1. The authors should specify the “4% histiocytic fixative” used to fix fertilized eggs.

Response: Thanks for your suggestion, and we changed the words into “4% paraformaldehyde solution” in the revision.

  1. The authors should define all the abbreviations at the time of the first appearance.

Response: Thanks for your reminding, and we have defined all the abbreviations at the time of the first appearance as you suggested.

  1. L279–282 and L286–288 require references.

Response: Thanks for your suggestion, and we added references for L279–282 in the position of the manuscript as you suggested in the revision. Acutally, the sentence L286–288 is one of the deductions according to our results and we have modified the statement properly in the discussion.

Round 2

Reviewer 1 Report

The authors have answered to most of the comments and improved the figures significantly.

It is however a pity that the new title shows a big mistake with "epididymitis" instead of "epididymis". In vitro should also always to written in 2 words.

Author Response

Specific response to Reviewer 1

Question: It is however a pity that the new title shows a big mistake with "epididymitis" instead of "epididymis". In vitro should also always to written in 2 words.

Reponse: Thanks for your kind reminding. We have modified "epididymitis" into "epididymis" and written “in vitro” properly in the revision.

Reviewer 2 Report

Thank you for revisions according to the comments. Most of the comments are satisfactorily addressed, but there are some points to be revised. Please consider the following comments.

  1. Distribution of F-actin in the epididymal epithelium appears not to be different (Fig. 5A). The genes examined in Fig. 4, such as Tubα3a, are downregulated, but a-tubulin shown in Fig. 5A appears to be upregulated. Please verify and discuss this discrepancy. This point is critical for the conclusion of this study.
  2. Revisions for comment #7 are still short. The authors should discuss what is caused by the deletion of Plz1 more specifically. As the authors indicated, Plz1 deletion models have also been reported elsewhere. Please discuss what function is estimated in the deleted region of Plz1m3 in this study, e.g., to spermatogenesis in the seminiferous tubule and maintenance of cytoskeleton in epididymal epithelium, both of which may affect sperm quality.
  3. “Spermatogenesis” is normally used for testicular function, and therefore, “sperm maturation” may be better in referring to a function of epididymis. 
  4. The authors should indicate the region of the epididymis examined because each segment of the epididymis shows different characteristics.
  5. GraphPad prism 8 (L182) is not indicated in the Materials and methods.

Author Response

Specific response to reviewer #2:

Question 1: Distribution of F-actin in the epididymal epithelium appears not to be different (Fig. 5A). The genes examined in Fig. 4, such as Tubα3a, are downregulated, but a-tubulin shown in Fig. 5A appears to be upregulated. Please verify and discuss this discrepancy. This point is critical for the conclusion of this study.

Reponse: Thanks for your question. In this study, abnormal structures were only found in epididymis of Plcz1m3 mice according to the results of Fig. 3 and Fig. S1, After RNA-seq, cytoskeleton changes in epididymis of Plcz1m3 mice were explored. Distributions of F-actin and a-tubulin was observed under fluorescence microscope. As shown in Fig. 5 A, F-actin and a-tubulin stained as green or red filaments was distributed over the whole cell in epididymal epithelium of WT but a-tubulin was mainly distributed around the plasma membrane to form a strong red fluorescence ring in the cell edge in epididymal epithelium of Plcz1m3. Actually, α-tubulin was disrupted in Plcz1m3. On one hand, accumulation of the irregular cells in epididymal epithelium makes it show stronger signals. It is improperly to for show the unrepresentative filed. In addition, the expression of α-tubulin protein may be not dependent on Tubα3a mRNA levels completely. Other mechanisms may also be involved in the disruptions of α-tubulin caused by Plcz1 deletion in Plcz1m3. To verify the results, we improved the quality of the picture to show it more clearly and supplied pictures from other samples in one group in Fig. S2 of supplementary materials. We are appreciated for the comments which are valuable for our further studies.

Question 2: Revisions for comment #7 are still short. The authors should discuss what is caused by the deletion of Plz1 more specifically. As the authors indicated, Plz1 deletion models have also been reported elsewhere. Please discuss what function is estimated in the deleted region of Plz1m3 in this study, e.g., to spermatogenesis in the seminiferous tubule and maintenance of cytoskeleton in epididymal epithelium, both of which may affect sperm quality.

Reponse: Thanks for your comments. On one hand, as a member of PLC family, Plcz1 can hydrolyze PIP2 to produce not only IP3 but also diacyl glycerol (DAG) in cells. DAG is one of the best characterized products of PLC mediated reactions and that is known to lead to downstream activation of serine/thereonine protein kinase C (PKC). Further, PKC signaling is also required for the protection against oxidant-induced cytoskeletal disruption (A. Banan etal. Am J Physiol Cell Physiol). In this study, the regulation of actin cytoskeleton is affected by Plcz1 deletion according to the KEGG results. Thus, we speculated that the deletion region of Plcz1 in Plcz1m3 can lead to the loss of its hydrolytic function and further caused the inhibition of PKC signaling, which further aggravated the disruption of cytoskeleton. However, the hydrolytic ability may be partly retained in Plcz1m5. In addition, Plcz1 deletion in Plcz1m3 but not Plcz1m5 may lead to the inhibition of the interaction with Capza3, which further contributed to the disruption of the cytoskeleton. Consistently, Mahmoud reported that Plcz1 is expressed and secreted by the epididymal epithelial cells during sperm maturation in the epididymis (Aarabi M, etal. Plos One), which indicated that Plcz1 played an important role in sperm maturation and the maintenance of epithelial physiology. Your comments are great for our further studies.

Question 3: “Spermatogenesis” is normally used for testicular function, and therefore, “sperm maturation” may be better in referring to a function of epididymis.

Reponse: We are appreciate for your valuable suggestion and we changed “spermatogenesis” into “sperm maturation” properly in the revision.

Question 4: The authors should indicate the region of the epididymis examined because each segment of the epididymis shows different characteristics.

Reponse: Thanks for your valuable suggestion. Actually, cauda epididymis was mainly examined in this study, which we have indicated in the revision.

Question 5: GraphPad prism 8 (L182) is not indicated in the Materials and methods.

Reponse: Thanks for your kind reminding and it has been indicated in the Materials and methods (L466) of the revision.
